# Development and Characterizations of Engineered Electrospun Bio-Based Polyurethane Containing Essential Oils

**DOI:** 10.3390/membranes12020209

**Published:** 2022-02-10

**Authors:** Nehir Arik, Nesrin Horzum, Yen Bach Truong

**Affiliations:** 1Department of Biocomposite Engineering Graduate Program, Izmir Katip Celebi University, Izmir 35620, Turkey; nhrark@gmail.com; 2Department of Engineering Sciences, Izmir Katip Celebi University, Izmir 35620, Turkey; 3CSIRO Manufacturing, Bag 10, Clayton, VIC 3168, Australia; yen.truong@csiro.au

**Keywords:** biomaterials, electrospinning, essential oils, nanofibers, polyurethane

## Abstract

We report the fabrication of bio-based thermoplastic polyurethane (TPU) fibrous scaffolds containing essential oils (EO). The main goal of this study was to investigate the effects of essential oil type (St. John’s Wort oil (SJWO), lavender oil (LO), and virgin olive oil (OO))/concentration on the electrospinnability of TPU. The effects of applied voltage, flow rate, and end-tip distance on the diameter, morphology, and wettability of the TPU/EO electrospun fibers were investigated. The electrospun TPU/EO scaffolds were characterized by scanning electron microscopy (SEM), atomic force microscopy (AFM), contact angle (CA), and Fourier transform infrared spectroscopy (FTIR). The addition of oil resulted in an increase in the fiber diameter, reduction in the surface roughness, and, accordingly, a reduction in the contact angle of the composite fibers. TPU fibers containing SJWO and LO have a more flexible structure compared to the fibers containing OO. This comparative study fills the existing information gap and shows the benefits of the fabrication of essential-oil-incorporated electrospun fiber with morphology and size range with respect to the desired applications, which are mostly wound dressing and food packaging.

## 1. Introduction

Electrospinning is a favorable and powerful technique to fabricate uniform sub-micron fibers in a continuous process and on a large scale from many different polymers. Electrospun fibers, from submicron down to nanometer, with a high surface area are produced from polymer melt or solution by electrostatic forces [1,2]. Various synthetic and natural polymers, composites, and ceramics have been transformed into fibers to benefit their surface area characteristics for several applications, such as filtration, catalysis, energy, sensor, food packaging, and healthcare. Green nanofibers that contain bioactive reagents, such as essential oils, herbal extracts, polyphenols, biopolymers, and some inorganic particles, have been designed to reduce the consumption of synthetic plastics [3,4].

Essential oils, which are native liquid oils, have several phytochemical components that are volatile and degrade quite swiftly due to the air, moisture, and temperature. For this reason, the controlled release of the essential oils can be achieved by electrospinning, encapsulation (micro/nano encapsulation, cyclodextrin inclusion complexes), multilayer system, nanoemulsion, and Pickering emulsion [5]. Recently, researchers have focused on the applications of electrospun fibers/essential oil composites in wound dressing, food packaging, bed net containing green insecticides, larvicidal bioassays, native anticancer, and antimicrobial drugs [6]. Four different strategies have been reported to obtain polymer/essential oil composite fibers: (i) essential oil that is blended with a polymer solution and subsequently electrospun, (ii) electrospinning from a polymer solution containing essesntial oil filled in the carrier, such as beta-cyclodextrin or its derivates, PVA microcapsules, and activated carbon, etc., (iii) essential oil (core)–polymer solution (shell) electrospinning [6], (iv) loading essential oil by imbibition onto the electrospun mat [7].

Argan, black pepper, candeia, cinnamon, clove, garlic, lavender, limonene, olive, orange, oregano, palmarosa, pomegranate seed, and tea tree oils were incorporated or encapsulated into the nanofibers from synthetic polymers (polyamidoamine (PAMAM), polycaprolactone (PCL), polyethylene oxide (PEO), polylactic acid (PLA), polyurethane (PU), polyvinyl alcohol (PVA), and polyvinyl pyrrolidone (PVP)), polysaccharides (alginate, cellulose, chitosan, dextran pullulan, and starch), and food proteins (collagen, gelatin, hordein, silk, and zein) [8,9,10,11]. Among these carriers, polyurethane (PU), which is an FDA-approved polymer, is often preferred as a material for forming a nanofibrous mat due to its superior properties, such as its chemical stability, mass transport, and excellent mechanical properties [12].

Electrospun PU mats have attracted attention in many fields, such as adhesives and coatings, elastomers, foams, and medical applications. The integration of biodegradable and biocompatible bio-based PU with the human skin over time is among the desired properties of the membrane, and its use as a wound dressing has become widespread [13]. Khil et al. [14] showed that the polyurethane membrane is suitable as a wound dressing due to its controlled fluid loss, excellent oxygen permeability, and the fact that it encourages the discharge of fluid due to the porosity of nanofibers. An ideal scaffold for biomedical applications, such as wound dressing and tissue engineering, should have an enhanced biocompatibility to affect the fibroblast adhesion, and should also have antimicrobial activity. Many studies have reported that natural products, such as essential oils and plant extracts, increase the cellular response. As a representative example, Unnithan et al. [15] fabricated electrospun PU/essential oil nanofibers. It has been reported that emu-oil-containing PU scaffolds can be preferred for therapeutic and cosmetic applications as they show an enhanced cell proliferation and cytocompatibility. Furthermore, it was observed that the scaffold incorporating metallic particles accelerated the wound healing process and showed an improvement in the tensile strength. Amina et al. [16] were probably the first to show the fabrication of electrospun nanofibers from a PU/virgin olive oil/silver nanoparticles blend as bionanocomposite scaffolds for wound dressing. By using a similar strategy, antibacterial and anti-inflammatory organic/inorganic hybrid micronanofibers composed of PU/virgin olive oil/copper oxide nanocrystals have been suggested for use in tissue regeneration and damaged skin treatment by Amna et al. [17]. In both studies, the optimal content of PU, olive oil, and nanomaterials was standardized as 10, 5, and 3 wt %, respectively, in order to obtain a spinnable solution for the fabrication of composite nanofibrous membranes.

Commercially available indhulekha oil and bio oil${}^{\mathrm{TM}}$ were incorporated into PU nanofibers, and anticoagulant properties of the potential skin patch were pointed out by activated partial thromboplastin and prothrombin time assays [18,19]. The same group of authors proved that the addition of grapeseed oil to PU nanofibers enhanced fibroblast adhesion and proliferation [20]. Inspired by their previous works, the advantages of essential oil and metallic particles were combined into PU/peppermint oil/${\mathrm{CuSO}}_{4}$ [21] PU/lavender oil/${\mathrm{CoNO}}_{3}$ [22] nanofiber scaffolds to be used for a blood coagulation retarder.

In a different work, PU/PMMA nanofibers were first fabricated and then treated with peppermint oil [7]. The contribution of the controlled release of peppermint oil from the nanofibrous membrane to prolonging the postharvest storage time of potatoes and preventing sprouting at room temperature was investigated. Another food packaging application to extend the shelf life of fruits was carried out using PU/cinnamon oil by Lee et al. [23]. Similarly, they used tea-tree-oil-infused PU nanofibers, characterized by mechanical, ${\mathrm{CO}}_{2}$ reduction, and antibacterial tests, to extend the shelf-life of tomatoes [24]. Besides tea tree oil, PU nanofibers containing cinnamon bark and clove oils were also fabricated, and the antimicrobial activities were examined [25]. Unlike polymer/essential oil blend electrospinning, they also attempted the adsorption of cinnamon oil on the activated carbon, then mixed it with a PU solution, and subsequently electrospun it to obtain a higher amount of oil in the composite nanofibers [26]. The air filtration efficiency and pressure drop of the produced antimicrobial nanofibers have been observed and reported to be applicable to various fields, such as personal masks, air purifiers, and biological applications. The most commonly used essential oil with PU nanofibers was cinnamon. The antimicrobial properties of PU/*Cinnamomum zeylanicum* against major burn wound pathogens were investigated for not only the treatment of burn wound infections but also for increasing wound healing and reducing antibiotic use [27].

In this work, TPU fibrous scaffolds containing EO were successfully fabricated. *Hypericum perforatum* L. (Hypericaceae), widely known as SJWO, is a medicinal flowering plant that is partly or completely involved in the treatment of anxiety and depression [28]. The components of SJWO, hyperforin, and hypericin are claimed to have anti-inflammatory, antiviral, and antimicrobial activities that help to reduce both the wound size and healing time [29,30]. LO is extracted from *Lavandula angustifolia*, and is a traditional plant extract that is used for minor burns and insect bites due to its anti-inflammatory and calming activity. It has many beneficial qualities, including reducing anxiety, acting as an antioxidant, having anticancer and anti-mutagenic properties, alleviating pain, and being suitable for the management of central nervous system disorders. Linalool and linalyl acetate are the two main ingredients of LO, though it has many other compounds. Many studies revealed that these main ingredients have fairly high anti-inflammatory activity, besides antibacterial and antifungal properties, toward *Staphylococcus aureus*, which is the bacterium mostly responsible for burn wound infections [31]. Oleuropein is the dominant phenolic compound in olive, although it is not found in OO; instead, the phenols in olive oil are elenolic acid, hydroxytyrosol, and tyrosol, besides vitamin E, which promote antioxidant activity to olive oil [32]. OO is commonly utilized in various areas, such as gastronomy, cosmetics, the pharmaceutical sector, and, moreover, as a fuel in kerosene lamps [33]. The main goal of this study was to investigate the effects of SJWO, LO, and OO and their concentration (1, 5, 10, 15, 20, and 25 wt %) on the electrospinnability of TPU. The morphology, wettability, and chemical structure of nanofibers were evaluated in different system parameters, such as applied voltage, flow rate, and end-tip–collector distance. Our research showed the integration of essential oils, SJWO, LO, or OO to the TPU nanofiber and those composite nanofibers hold great promise for several applications, such as wound dressing, food packaging, and air filtration.

## 2. Materials and Methods

### 2.1. Materials

Tecophilic${}^{\mathrm{TM}}$ TPU was purchased from the Pearlthane^®^ ECO D12T85, Lubrizol Company and was used as raw material for biopolymer matrix, and *N, N*-dimethyl formamide (DMF, Sigma-Aldrich, St. Louis, MO, USA) was used as a solvent. SJWO and OO were received from Saracoglu Incorporated Company in Turkey. LO (61718 natural, from *Lavandula angustifolia* L.) was supplied by Sigma-Aldrich (Sigma-Aldrich, St. Louis, MO, USA). Figure 1 demonstrates the chemical structure of the polymer and main components in EOs which are employed in this study. All samples were used without further purification.

### 2.2. Fabrication of TPU/EO Scaffolds

Preparation solution and fabrication of TPU nanofiber membrane was performed as previously mentioned [34]. In brief, the polymer solutions were prepared by dissolving the PU in DMF at the concentrations of 5, 10, 12.5, and 15 wt % and leaving for 24 h under magnetic stirring at approximately 25–30 °C. The processing parameters were varied to evaluate their effect on fiber formation and morphology: a DC voltage of 10.0–12.5–15.0 kV potential was applied between the syringe tip and an aluminum sheet–collector at a distance of 14–17–20 cm and a syringe flow rate of 1.00–1.25–1.50 mL h^−1^ at an ambient temperature of 25 °C.

Provided that the TPU concentration (12.5 wt %) remains constant, EOs were added separately into the TPU solution at different concentrations (1, 5, 10, 15, 20, 25 wt %), then left for 24 h under magnetic stirring. Table 1 summarizes the all-solution concentrations obtained for each essential oil. After TPU/EO blend solution became transparent, the nanofibers were fabricated using a commercial electrospinning platform (Inovenso Basic Setup) covered with a polycarbonate cabinet for safety and to avoid air convection. Electrospinning working parameters: a DC applied voltage that was 10.0 and 15.0 kV in a gradient of 2.5 kV; flow rate was in the range of 1.0 and 1.5 mL h^−1^ in a gradient of 0.25 mL h^−1^, and end-tip-to-collector distance ranged between 14 and 20 cm in a gradient of 3 cm at an ambient temperature of 25 °C. The optimum values of the electrospinning parameters that were selected for each solution were 12.5 kV, 1.0 mL h^−1^, and 17 cm.

### 2.3. Characterization TPU/EO Scaffolds

The surface properties and morphology of the TPU and essential oil-loaded TPU groups were investigated using scanning electron microscopy (SEM; Carl Zeiss 300 VP) operating at an accelerating voltage of 5 kV. Prior to the imaging, a gold coating was applied to the fibrous samples using a Q150 RES coater to enhance the conductivity. On SEM micrographs, at least 100 fiber segments were randomly selected, and the average diameter was measured through ImageJ software.

The conductivity of the solutions was measured by an electrical conductivity meter (Inolab-IDS Multi 9310). The temperature was adjusted to room temperature before the measurements and viscosity of the TPU and EO/TPU solutions were obtained by the viscometer at 20.00 rpm (Thermo scientific HAAKE Viscositester C). The wettability of the pure and oil–loaded PU scaffolds were measured via a contact angle measuring device (CA; Theta Lite Optical Tensiometer, Finland). The surface roughness parameters were measured from the images taken by atomic force microscope (AFM) (Nanosurf AG, Easyscan-2, Liestal, Switzerland).

The chemical modification of TPU due to blending with essential oils (SJWO, LO, OO) was characterized using Fourier transform infrared spectrometry (FTIR; Thermo Scientific${}^{\mathrm{TM}}$ Nicolet${}^{\mathrm{TM}}$ iS${}^{\mathrm{TM}}$5, Waltham, USA).

## 3. Results and Discussion

TPU fibers containing three different essential oils (SJWO, LO, and OO) were fabricated by electrospinning (Figure 2). The optimization of fiber fabrication depends on various factors, including solution, process, and environmental parameters. The effect of polymer or essential oil amount, applied voltage, flow rate, and end-tip-to-collector distance on the properties of TPU fibrous mats, such as morphology, hydrophilicity, and structure, was evaluated. Hydrophilic polyester-based thermoplastic aliphatic polyurethane is a medical-grade polymer that can absorb more than 150% of its dry resin weight [35]. In this way, the design of a TPU-based nanofiber membrane that provides constant humidity is envisaged in order to be used in biomedical applications.

### 3.1. Morphology of Electrospun TPU/Essential Oil Fibrous Membranes

In our previous study, we examined the effects of TPU solutions in different percentage concentrations and system parameters (voltage, flow rate, and end-tip–collector distance) on the fiber morphology [34]. A low polymer concentration caused beaded fibrous structures; however, the bead density decreased with an increasing polymer concentration. This was due to the fact that the viscosity of the solution increased with an increasing TPU concentration, and that the solution dispersed in the generated electric field, causing it to accumulate continuously on the collector. Another reason for the increase in viscosity was the high molecular weight of the polymer used [36]. Thus, it caused the polymer chains in the solution to stick together, increasing the viscosity of the solution. The diameters of the nanofibers increased as the polymer concentration increased. The reason for this change can be related to the increase in viscosity [35]. The viscosity of TPU/DMF solutions with different concentrations (5, 10, 12.5, and 15 wt %) were 66.8, 136.2, 438.9, and 588.1 mPa·s, respectively (Table S1). As the viscosity increases, the path of the jet decreased, so the solution was less elongated and the diameters of the fibers increased. As mentioned before [34], the diameters of the nanofibers obtained from 5, 10, 12.5, and 15 wt % TPU solutions are 250 ± 80 nm, 370 ± 130 nm, 480 ± 150 nm, and 850 ± 250 nm, respectively. Table 2 indicates the viscosity and conductivity of TPU solutions containing essential oils that resulted in a higher viscosity and lower conductivity. Essential-oil-free TPU has the highest conductivity of 2.0 µS cm^−1^, and the conductivity values relatively decreased with the addition of essential oils. SJWO and LO have no significant influence on the conductivity of solutions, whereas the conductivity decreased with an increasing OO concentration. The change in viscosity and conductivity can be correlated with the fiber diameter. The presence of an increasing concentration of essential oil in polymer solutions means a generally high viscosity, low surface tension, and conductivity; therefore, it leads to an insufficient stretching of the electrified jet during electrospinning, and more uniform thicker fibers are obtained [37,38].

Figure 3 shows the SEM micrographs (a–f) and diameter distribution (g) of the electrospun fibers of the different formulations, as well as the photographic image of the TPU/SJWO mat (h). The largest diameter belongs to the fiber containing 10 wt % SJWO.

A significant increase in diameter was observed with the addition of SJWO. The reason for this may be the encapsulation of the oil droplets in the fibers and the thickening of the fiber. When the SJWO concentration was increased from 1 to 10 wt %, the fiber diameter of TPU/SJWO gradually increased (440 ± 60 nm, 915 ± 105 nm, 1870 ± 245 nm for 1, 5, and 10 wt %, respectively); however, for the oil concentrations above 10 wt %, the diameter fluctuated in themselves, although they have a larger diameter than pristine TPU fibers. The diameters of the TPU fibers containing 15, 20, and 25 wt % SJWO were 1460 ± 185 nm, 1035 ± 180 nm, and 1360 ± 240 nm, respectively. As high concentrations were reached, oil residues were also observed on the aluminum foil separately from the fibers. This may be due to the fact that the amount of solvent in solutions containing high concentrations of SJWO are insufficient, that the oil becomes insoluble in the solution, that all of them cannot be converted to fiber because of the low volatility of SJWO, and that it cannot dry until reaching the collector.

Table 3 presents the average fiber diameter of TPU/SJWO fibers obtained at different electrospinning conditions, such as applied voltage, flow rate, and end-tip–collector distance. A decrease in the applied voltage resulted in a decrease in the fiber diameter. The reason why the fiber diameter is thinner under the effect of a low voltage is that the voltage increases the flight time of the electrospinning jet by creating a slow jet velocity and weak electric field, thus resulting in the formation of finer fibers [39]. When the flow rate and the end-tip-distance were increased, the diameter of the resulting TPU/SJWO fibers increased. Due to the charging effect from the electron beam during SEM imaging, the fibers obtained at 15.0 kV, 1.50 mL h^−1^, and 20 cm cannot be imaged properly to calculate the fiber diameters.

SEM micrographs of the LO-loaded TPU fibers are shown in Figure 4a–f. The surface of the fibers formed generally have a smooth appearance. However, deformations are noticeable in some parts of the fibers obtained from the TPU solution containing 5 wt % LO. The reason might be the gold plating prior to imaging to improve the conductivity. The fibers obtained from the solution containing 20 and 25 wt % LO had no integrity and adhered to each other because of the high concentration of the oil.

Figure 4g shows the diameter changes in concentration-dependent TPU/LO fibers. An increase in diameter was observed with the addition of LO to a pristine TPU. These results showed that LO penetrates the fibers and increases the diameter. However, the diameter of TPU nanofibers containing 25 wt % LO (390 ± 60 nm) was smaller than that of the LO-free fiber (480 ± 150 nm). This might be due to the higher concentration of the oil, since the evaporation was low and did not penetrate the fiber, thereby reducing the diameter. The nanofiber diameters obtained from solutions containing 1, 5, 10, 15, and 20 wt % LO are 640 ± 125 nm, 595 ± 115 nm, 475 ± 115 nm, 600 ± 105, and 520 ± 90 nm, respectively.

The physical appearance of the TPU/LO fibrous mat is shown in Figure 4h. TPU fibers containing SJWO and LO have a flexible structure. The optimization electrospinning parameter was continued with TPU fibers containing 20 wt % LO. Table 4 lists the average fiber diameter of the TPU/LO fibers as a function of the applied voltage, flow rate, and end-tip–collector distance. The largest diameter belongs to the fiber obtained by applying a voltage of 12.5 kV. No significant difference was observed in the diameters of fibers with the increasing flow rate. Similarly, a certain increase/decrease trend between the diameters could not be obtained by changing the end-tip-distance.

SEM micrographs of TPU/OO fibers obtained from different concentrations of OO, which is classified more accurately as a carrier oil, that were used in this study are given in Figure 5a–d. Although TPU fibers containing OO were successfully fabricated for all concentrations, the fibers having 15 and 25 wt % OO could not be imaged properly to calculate the fiber diameters because of the charging effect from the electron beam during SEM imaging. As the OO concentration increased, thicker fibers were obtained (Figure 5e) and the yellow-colored robust fibrous mat was formed (Figure 5f). Although there is an increasing trend of fiber diameter as a function of OO containing TPU fibers, no clear trend in the diameter of SJWO and LO containing fibers was observed. This might be explained by the difference in the thickness and porosity of the imaged parts. In general, according to the fiber diameter screening, it was determined that the TPU fibers containing SJWO were the largest, and that the TPU fibers containing LO were the thinnest.

### 3.2. Wettability

The contact angle and surface energy of the TPU fibers containing different concentrations of essential oils are shown in Figure 6. It can be seen that, with an increased oil concentration, the contact angles decrease, rendering them less hydrophobic. Figure 6a shows that the contact angle of the pristine TPU fibers is 94 ± 5.0°, whereas the contact angles of the TPU fiber with increasing SJWO concentrations (1, 5, 10, 15, 20, and 25 wt %) are 95 ± 1.5°, 80 ± 8.0°, 65 ± 9.0°, 65 ± 6.0°, 43 ± 2.0°, and 62 ± 8.5°, respectively. The reason for this decrease might be that the amount of added essential oil reduces the solid–liquid interface voltage by adsorbing on the solid surface [40].

Table 5 shows the changes in the contact angle and surface energy versus system parameters applied to the TPU/SJWO (20 wt %) fiber membrane. The diameter of the fiber increased with the increase in the applied voltage, and more solutions from the syringes were ejected. Therefore, the higher amount of oil was trapped in the fiber and reduced the contact angle by increasing the hydrophilicity. However, when the voltage amount was increased to 15 kV, it was observed that the contact angle increased. At a high voltage, the solutions might not be completely transformed into a fiber, and polymer agglomerates might also be present within the mesh; additionally, the surface is rough. As the flow rate increased, the contact angle of the fibers showed an increasing trend. In this case, it can be argued that the surface tension increases with an increasing flow rate.

When the end-tip distance is 14 cm, the highest value of the contact angle is measured. The SJWO amount in the solution might not be able to evaporate in a short distance. Therefore, the amount of SJWO in the fiber accumulated on the collector is very low. With an increasing distance (17 cm), the amount of oil trapped in the fiber increased, thereby increasing its hydrophilicity. The contact angle of the fiber obtained when the distance is 20 cm may be due to the fact that the solution could not properly reach the collector. In other words, the contact angle value may have increased due to the low amount of oil in the fiber.

The contact angle and surface energy values of TPU fibers obtained from different concentrations of LO are given in Figure 6b. It has been shown that, with an increasing LO concentration, the contact angles of the fibers decrease; that is, the surface wettability properties increase. Similar to SJWO, this decreasing trend can be interpreted as increasing the surface energy of the fiber obtained by increasing the amount of oil in the solution, thereby increasing its wettability [41]. Although the contact angle showed a decreasing trend with the increase in the amount of LO in the solution, the contact angle value was found to be 90 ± 0.1° at an intermediate value of 20 wt %. This may be due to the surface being rougher than the other concentrations, or the surface energy being low.

Table 6 shows the changes in the contact angle and surface energy values in response to the system parameters applied to TPU nanofibers containing 20 wt % LO. On one hand, when the applied voltage is increased from 10.0 kV to 12.5 kV, it is observed that the contact angle value increases; however, when it is increased from 12.5 kV to 15.0 kV, it did not increase more than this value. On the other hand, changing the flow rate and end-tip distance has no major influence on the contact angle values.

Figure 6c shows the wettability of TPU/OO fibers at different OO concentrations. Similarly, the increase in OO concentrations (1, 5, 10, 15, 20, and 25 wt %) in the nanofiber results in a decrease in the contact angle values (90 ± 4.0°, 88 ± 1.3°, 81 ± 1.2°, 75 ± 0.4°, and 77 ± 1.5°). TPU/OO fibers are observed to be less hydrophobic than pristine TPU fibers.

In general, the effect of decreased contact angles is shown with a decreased fiber diameter [42]. The thickest TPU/SJWO fiber has the lowest contact angle. The wetting of the fibers can be examined not only by the average fiber diameter but also by the surface roughness and fiber fraction, given the identical fiber thicknesses keep the electrospinning time constant [43]. The representative AFM images and surface roughness parameters of TPU/EO fibers are given in Supplementary Materials (Figure S1 and Table S2). The roughest surface belongs to the TPU fiber; the addition of oil reduces the roughness, so the contact angle decreases, whereas the surface energy increases.

### 3.3. Chemical Structure Analysis

The FTIR spectra of TPU fibers containing different concentrations of EOs are given in Figure 7. For comparison, FTIR spectra of TPU (bio-based) and PU (commercial) fibers are also examined (Figure S2). The characteristic absorption bands of the TPU fibers observed are C =O (amide I) stretching at 1729 cm${}^{-1}$, O–C–N (amide II) stretching at 1530–1595 cm${}^{-1}$, N–H (amide III) bending at 1260 cm${}^{-1}$, and N–H stretching vibrations at 3330 cm${}^{-1}$. Other characteristic bands were also observed in alkane –CH_2_ stretching at 2927–2846 cm${}^{-1}$, C–N and C–O stretching at 1226 cm${}^{-1}$, and ester C–O–C symmetrical stretching vibrations at 1166 cm${}^{-1}$[44]. Figure 7a shows the suppression of the N–H stretching band of TPU nanofibers at 3319 cm${}^{-1}$ by the addition of SJWO. The intensity of C=O (amide I) stretching at 1729 cm${}^{-1}$ increases as the concentration of SJWO in nanofiber content increases. This was due to the phenolic compounds in SJWO [45]. Similarly, N–H (amide III) bending at 1260 cm${}^{-1}$, C–N and C–O stretching vibrations at 1226 cm${}^{-1}$, and ester C–O–C symmetric stretching at 1166 cm${}^{-1}$ are directly proportional to the increase in the concentration of SJWO in the fiber.

Figure 7b shows the functional groups present in TPU/LO fibers obtained from different concentrations of LO. O–C–N (amide II) stretching at 1530–1595 cm${}^{-1}$ and C–N and C–O stretching vibrations [46] were found to be suppressed in low concentrations of LO content. Lafhal et al. [47] have reported the characteristic medium-IR (MIR) spectrum for LO in the 1800–1650 cm${}^{-1}$. The carbonyl group of lavendulyl acetate, camphor, linalyl acetate, and 3–octananone was reported at 1740, 1738, 1736, and 1713 cm${}^{-1}$, respectively. Additionally, the C=C band of lavendulyl acetate, linalyl acetate, linalool, and β-cariophylene appeared at 1647, 1645, 1640, and 1633 cm${}^{-1}$, respectively. However, it can be interpreted that, as the LO concentration increases, the intensity of the vibrations increases, since these functional groups are also present in TPU.

FTIR spectra of TPU/OO fibers containing different concentrations of OO are presented in Figure 7c. The absorption band at 3319 cm${}^{-1}$ is the characteristic N–H stretching of the TPU, suppressed by the addition of OO. An increase was observed both in the CH_2_ at 2927–2846 cm${}^{-1}$ and in the C=C band at 1666 cm${}^{-1}$, with an increasing OO concentration indicating the incorporation of OO in the composite fiber [48].

## 4. Conclusions

The feasibility of fabricating electrospun TPU fibers containing three different essential oils (SJWO, LO, and OO) has been demonstrated for the first time, to the best of our knowledge. This study is a pioneering work to fabricate truly green fibers using biodegradable polymers, non-toxic solvents, and bioactive agents for applications beneficial to human and public health. The diameters of fibers increased, whereas the contact angle values decreased upon the addition of oils. The order of biocomposite fibers according to their diameters was TPU < TPU/LO < TPU/OO < TPU/SJWO. On one hand, TPU/SJWO fibers became more hydrophilic and had a larger diameter of fibers than pristine TPU scaffolds with an increasing SJWO concentration in the blend. On the other hand, TPU/LO and TPU/OO fibers, which also have larger diameters than that of pristine TPU fibers, remained hydrophilic in all concentrations. The data obtained from this study will be focused on supporting the fabricated fiber scaffolds with in vitro studies by performing antimicrobial activity, controlled drug release, cell proliferation, and cytotoxicity tests. It is anticipated that the use of biocomposite nanofiber membranes in many areas, including biomedical engineering, material engineering, and healthcare, will be promising. We expect that this study will also contribute to the progressive development of essential oils in smart food packaging applications.

## Figures and Tables

**Figure 1 membranes-12-00209-f001:**
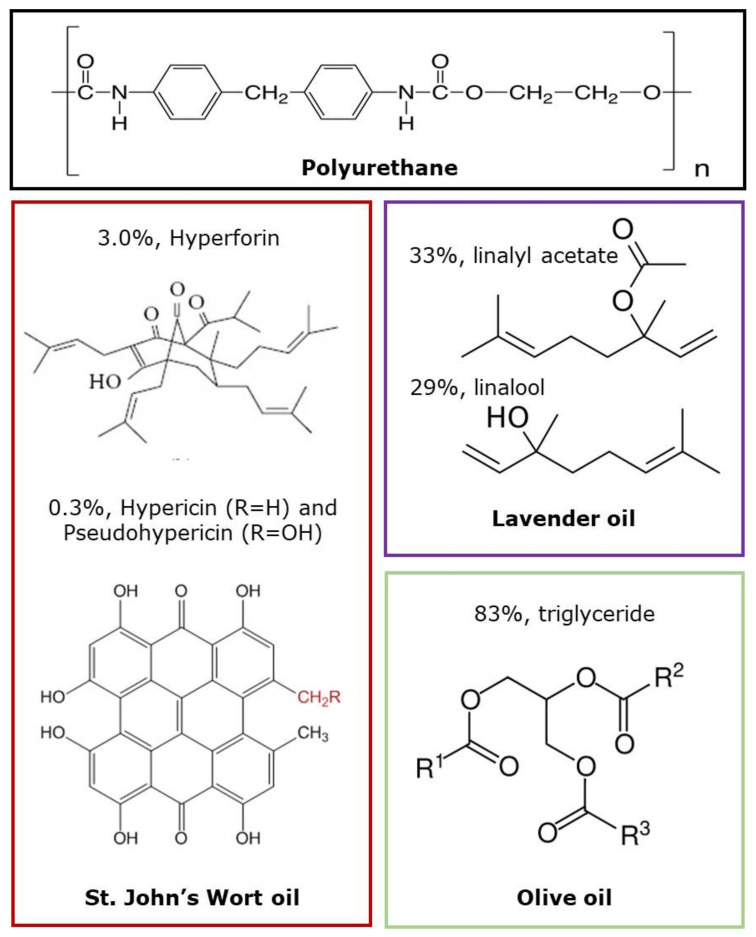
Chemical structure of the polymer and the main components in EOs used in this study.

**Figure 2 membranes-12-00209-f002:**
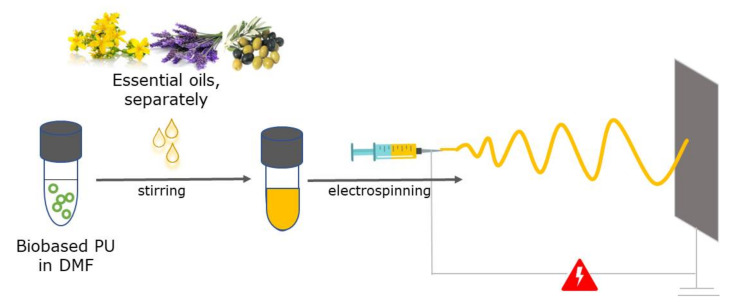
Schematic representation of TPU/EO composite fiber fabrication.

**Figure 3 membranes-12-00209-f003:**
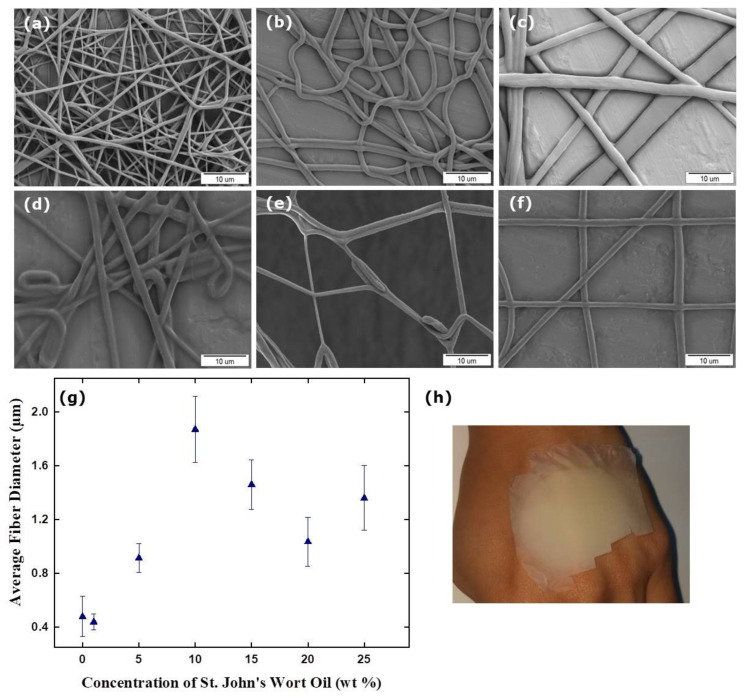
SEM micrographs with different concentrations of SJWO: (**a**) 1 wt %, (**b**) 5 wt %, (**c**) 10 wt %, (**d**) 15 wt %, (**e**) 20 wt %, (**f**) 25 wt %, the diameter distribution histogram (**g**), and photograph (**h**) of the TPU/SJWO fibers (applied voltage: 12.5 kV, flow rate: 1.00 mL h^−1^, distance: 17 cm, and TPU concentration in TPU/SJWO fibers: 12.5 wt %).

**Figure 4 membranes-12-00209-f004:**
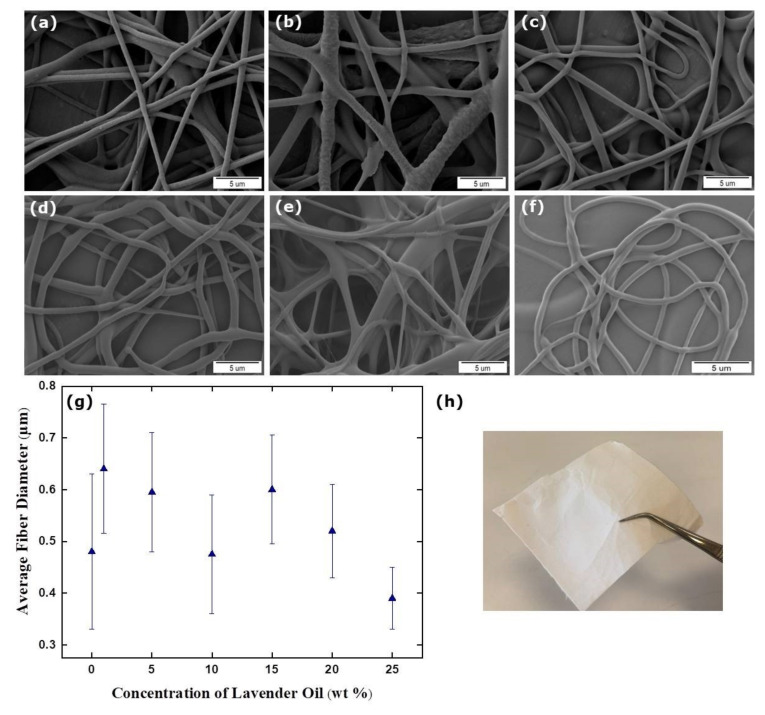
SEM micrographs with different concentrations of LO: (**a**) 1 wt %, (**b**) 5 wt %, (**c**) 10 wt %, (**d**) 15 wt %, (**e**) 20 wt %, (**f**) 25 wt %, the diameter distribution histogram (**g**), and photograph (**h**) of the TPU/LO fibers (applied voltage: 12.5 kV, flow rate: 1.00 mL h^−1^, distance: 17 cm, and TPU concentration in TPU/LO fibers: 12.5 wt %).

**Figure 5 membranes-12-00209-f005:**
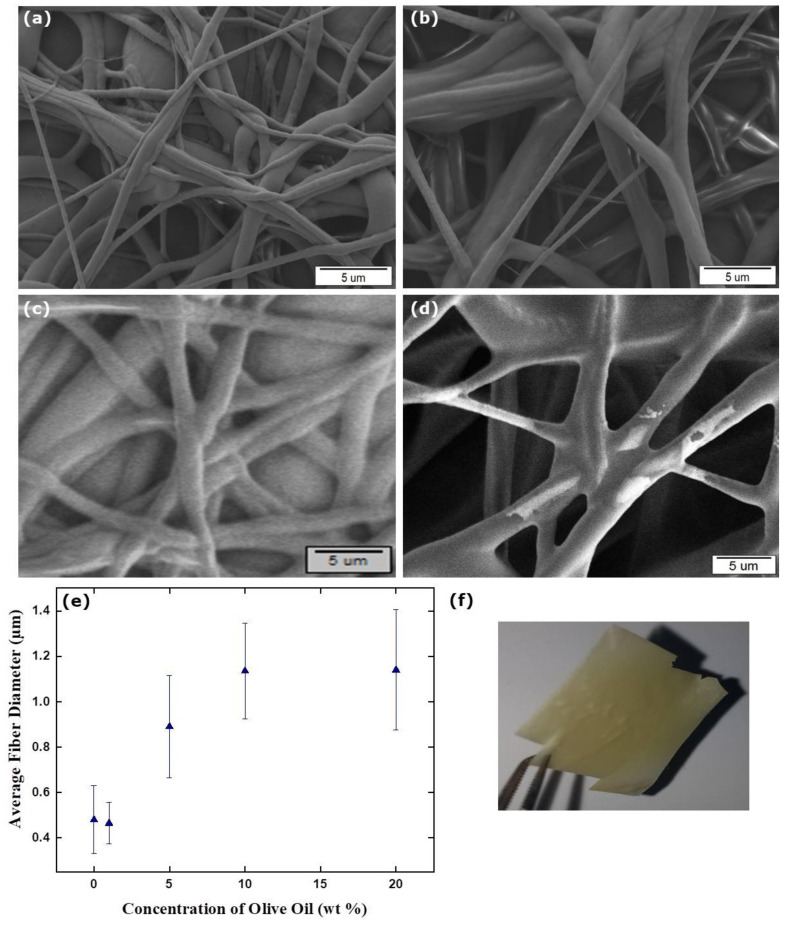
SEM micrographs with different concentrations of OO: (**a**) 1 wt %, (**b**) 5 wt %, (**c**) 10 wt %, (**d**) 20 wt %, the diameter distribution histogram (**e**), and photograph (**f**) of the TPU/OO fibers (applied voltage: 12.5 kV, flow rate: 1.00 mL h^−1^, distance: 17 cm, and TPU concentration in TPU/OO fibers: 12.5 wt %).

**Figure 6 membranes-12-00209-f006:**
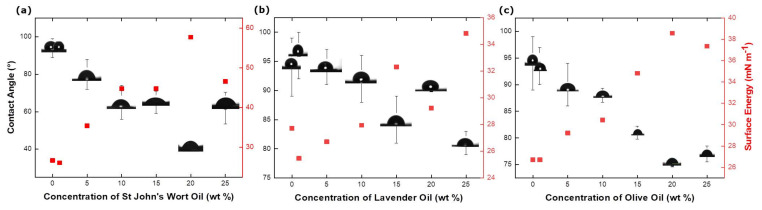
Contact angle and surface energy values of (**a**) TPU/SJWO fibers, (**b**) TPU/LO fibers, and (**c**) TPU/OO fibers (applied voltage: 12.5 kV, flow rate: 1.00 mL h^−1^, distance: 17 cm, and TPU concentration in the solutions: 12.5 wt %).

**Figure 7 membranes-12-00209-f007:**
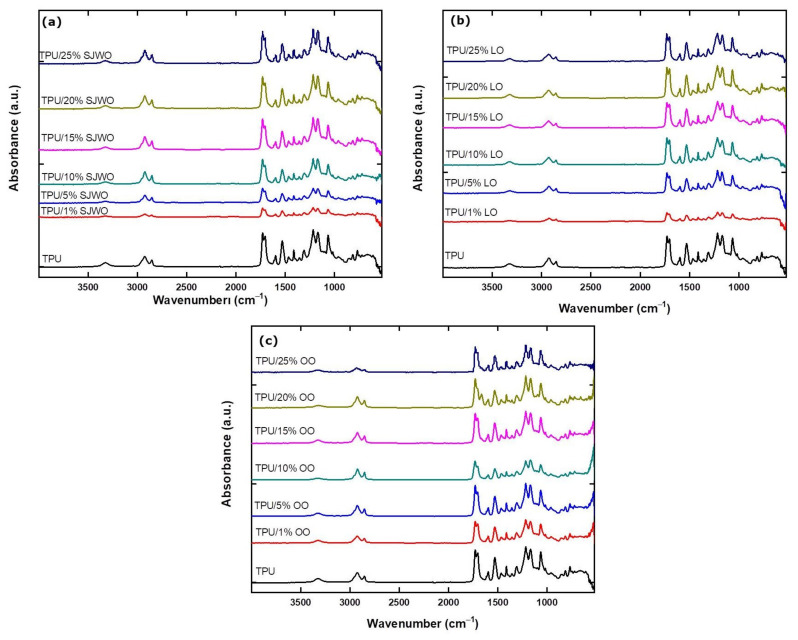
FTIR spectra of (**a**) TPU/SJWO fibers, (**b**) TPU/LO fibers, and (**c**) TPU/OO fibers.

**Table 1 membranes-12-00209-t001:** The concentrations of the solutions and system parameter values.

TPU Concentration in DMF (wt %)	Essential Oil Type	Essential Oil Concentration (wt %)	Electrospinning Parameters		
			**Applied Voltage** **(kV)**	**Flow Rate** **(mL h^−1^)**	**End-Tip-to-Collector** **Distance (cm)**
12.5	SJWO	1–5–10–15–20–25	12.5	1.00	17
	LO				
	OO				
	SJWO	20	10.0–12.5–15.0	1.00–1.25–1.50	14–17–20
	LO				
	OO				

**Table 2 membranes-12-00209-t002:** Characteristic properties of TPU/essential oil solutions: electric conductivity (σ), viscosity (µ); (12.5 wt % TPU (σ): 2.0 µS cm^−1^ (µ): 439 mPa·s).

EO Concentration (wt %) in 12.5 % TPU	TPU/SJWO		TPU/LO		TPU/OO	
	µ (mPa·s)	σ (µS cm^−1^)	µ (mPa·s)	σ (µS cm^−1^)	µ (mPa·s)	σ (µS cm^−1^)
1	1178	1.8	1040	1.6	960	1.8
5	1058	1.6	1105	1.6	1358	1.6
10	1191	1.6	1165	1.7	962	1.4
15	1096	1.6	1157	1.7	963	1.4
20	1168	1.6	1134	1.7	964	1.3
25	1067	1.6	1094	1.7	965	1.2

**Table 3 membranes-12-00209-t003:** The effect of system parameters applied for the fabrication of TPU fibers containing 20 wt % SJWO on the diameter.

System Parameters			Average Fiber Diameter (nm)
**Applied voltage** **(kV)**	10.0	Flow rate: 1.00 mL h^−1^	975 ± 155
	12.5	Distance: 17 cm	1035 ± 180
**Flow rate** **(mL h^−1^)**	1.00	Applied voltage: 12.5 kV	1035 ± 180
	1.25	Distance: 17 cm	1500 ± 495
**End-tip–collector** **(cm)**	14	Applied voltage: 12.5 kV	620 ± 160
	17	Flow rate: 1.00 mL h^−1^	1035 ± 180

**Table 4 membranes-12-00209-t004:** The effect of system parameters applied for the fabrication of TPU fibers containing 20 wt % LO on the fiber diameter.

System Parameters			Fiber Diameter Distributions (nm)
**Applied** **voltage (kV)**	10.0	Flow rate: 1.00 mL h^−1^	390 ± 160
	12.5	Distance: 17 cm	520 ± 90
	15.0		325 ± 65
**Flow rate (mL h^−1^)**	1.00	Applied voltage: 12.5 kV	520 ± 90
	1.25	Distance: 17 cm	560 ± 160
**End-tip–collector** **(cm)**	14	Applied voltage: 12.5 kV	440 ± 120
	17	Flow rate: 1.00 mL h^−1^	520 ± 90
	20		455 ± 135

**Table 5 membranes-12-00209-t005:** The effect of system parameters applied for the fabrication of TPU fibers containing 20 wt % SJWO on the contact angle and surface energy of the fibers.

Parameters			Contact Angle (°)	Surface Energy (mN m^−1^)
**Applied voltage (kV)**	10.0	Flow rate: 1.00 mL h^−1^	87 ± 3.0	31.10
	12.5	End-tip–collector: 17 cm	43 ± 2.0	57.73
	15.0		72 ± 0.3	40.47
**Flow rate (mL h^−1^)**	1.00	Applied voltage: 12.5 kV	43 ± 2.0	57.73
	1.25	End-tip–collector: 17 cm	64 ± 0.4	45.37
	1.50		82 ± 1.5	34.20
**End-tip–collector (cm)**	14	Applied voltage: 12.5 kV	95 ± 0.9	26.12
	17	Flow rate: 1.00 mL h^−1^	43 ± 2.0	57.73
	20		70 ± 2.0	41.71

**Table 6 membranes-12-00209-t006:** The effect of system parameters applied for the fabrication of TPU fibers containing 20 wt % LO on the contact angle and surface energy of the fibers.

Parameters			Contact Angle (°)	Surface Energy (mN m^−1^)
**Applied voltage (kV)**	10.0	Flow rate: 1.00 mL h^−1^	77 ± 4.0	37.36
	12.5	End-tip–collector: 17 cm	90 ± 0.1	29.22
	15.0		90 ± 0.5	29.22
**Flow rate (mL h^−1^)**	1.00	Applied voltage: 12.5 kV	90 ± 0.1	29.22
	1.25	End-tip–collector: 17 cm	91 ± 0.4	28.60
	1.50		83 ± 0.1	33.57
**End-tip–collector (cm)**	14	Applied voltage: 12.5 kV	90 ± 0.3	29.22
	17	Flow rate: 1.00 mL h^−1^	90 ± 0.1	29.22
	20		90 ± 0.6	29.22

## Data Availability

The data are contained within the article or Supplementary Materials.

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
