# Peer review of "Development and Characterizations of Engineered Electrospun Bio-Based Polyurethane Containing Essential Oils"

_membranes, 2022, doi:10.3390/membranes12020209_

Round 1

Reviewer 1 Report

Dear authors,

I read your paper and here are my comments:

The work has value and is interesting with good application potential.

  1. Introduction: "In this work, bio-based tecophilic polyurethane (TPU) fibrous scaffolds containing essential oils (EO)". This sentence does not have a meaning, please rephrase it.
  2. Which is the novelty of this paper with respect to similar papers in literature? This must be showed.
  3. Did you coat the samples with a conductive layer for SEM analysis? Gold coating or something similar.
  4. The first part of results section is too much literature. This could be moved in the introduction or added in the discussion part together with the obtained results.
  5. Please show the values for the free surface energies in contact angle measurements.
  6. What about the mechanical properties ot these electrospun materials?

Reviewer 2 Report

The paper presents research on the development and characterizations of engineered electrospun bio-based polyurethane containing essential oils. The current form's presentation of methods and scientific results is satisfactory for publication in the Membranes journal. The minor and significant drawbacks to be addressed can be specified as follows:
1.    The authors must supply an ORCID ID for all authors. Getting an ORCID iD is FREE, quick and easy to do through the ORCID registration page: https://orcid.org/register
2.    Lines 81 and 82. Park and Kim et al. ---> Lee et al. The first author of [24] is also Lee.
3.    Line 110, Prof Saracoglu. Please, give additional information: university, spin-off, company, web-page? https://www.profsaracoglu.com/ ???
4.    Tab. 1. (i) TPU Concentration ---> TPU concentration (ii) Applied Voltage ---> Applied voltage. See Tabs. 2 and 3. Please, standarize names.
5.    Lines 148 and 149, “Flow rateTecophilic polyurethane (TPU) fibers containing three different essential oils (St. John’s Wort oil (SJWO), lavender oil (LO), and olive oil (OO)”. See also line 172 and (OO). Summing up, Some abbreviations are explained several times in the text. It is sufficient to do only the first time.
6.    Fig. 5. Contact angle ---> Contact Angle.
7.    Fig. 4. (i) Why are SEM micrographs not shown for all concentrations of OO similar to the two previous figures? (ii) Fig. 4(a). It should be the same scale (y-axis) as in (b) and (c) panels.
8.    Fig. 6, panels (b) and (c). They should appear earlier in the manuscript. See also Figs. 7(b) and 8(b).
9.    Supplementary. Yen bach Truong ---> Yen Bach Truong
10.    Tabel S2. Where did the CA values come from? I was looking for such values in the text, and unfortunately, I did not find them. 
11.    Literature should also be standardized: the size of letters in the titles of journals as well as the titles of articles.

Reviewer 3 Report

Review of membranes-1557204

  1. Tecophilic is a brand from Lubrizol company. Please add TM (trademark sign) besides Tecophilic, as Tecophilic TM.
  2. In Lubrizol’s website there are several types of this brand, with different product codes. What type and what code of TPU product do you use in this experiment? Please add these missing details in the manuscript. Note: the website can be accessed here https://www.lubrizol.com/Health/Medical/Polymers/Tecophilic-TPU   
  3. Please write all units in X Y-1 format, not X/Y format, because for the FTIR result in this manuscript, the wavenumber is commonly written with cm-1 unit, and it will be strange to use “/cm” (it is not wrong, but not common). So, please write in this format for the units mL h-1, µS cm-1, and other units not yet mentioned in this review report.
  4. Figures 2 (TPU/St. John’s wort oil) and 3 (TPU/lavender oil): Why there is no clear trend of fiber diameter as a function of essential oil added into the TPU mixture? In Figure 4 (TPU/olive oil), there is a trend of a plateau (although it is not stable, with wide error bars). Please kindly explain.
  5. Line 54-55: There is this statement “…to affect fibroblast adhesion and also have antimicrobial activity…”, but in this manuscript, no study on fibroblast cells (such as L-929 mouse fibroblasts, etc.), and no antimicrobial tests (using common microorganisms such as Escherichia coli, Staphylococcus aureus, etc.). Please kindly explain.
  6. Line 25-69: This paragraph is too long, please separate as 2-3 shorter paragraphs.
  7. Line 70: Why “bio oilTM” is written with a trademark TM sign? Is it registered to a company? Please add the detail of the company.
  8. Line 92: Scientific name must be written in italic, with uppercase letter for the genus.
  9. Line 110: Please add the details of Prof Saracoglu as the producer of olive oil. At least write also the city and country where the olive is farmed and its oil is produced.
  10. Line 226-246: This paragraph is too long, please separate as 2-3 shorter paragraphs.
  11. Figure 3h: The picture of the produced polymer is not clear, cannot be distinguished from the aluminum foil. Please take again the photo, like Figure 4f.
  12. Figure 5: Why there are fluctuating trends of contact angle (Figure 5a, 5b, and less fluctuating in Figure 5c)
  13. Line 252-274: This paragraph is too long, please separate into 2-3 shorter paragraphs.
  14. Line 353: Please do not start a sentence with “And”. Please revise.
  15. Line 376: Please write the name of the reference with proper uppercase letter(s), only at the beginning of the words (excluding “of”, “in”, etc.).
  16. Line 387-388: Please write the name of the reference with proper uppercase letter(s), only at the beginning of the words (excluding “of”, “in”, etc.).
  17. Line 408: Please write the name of the reference with proper uppercase letter(s), only at the beginning of the words (excluding “of”, “in”, etc.).
  18. Line 423: Please write the name of the reference with proper uppercase letter(s), only at the beginning of the words (excluding “of”, “in”, etc.).
  19. Line 427: Please write the name of the reference with proper uppercase letter(s), only at the beginning of the words (excluding “of”, “in”, etc.).
  20. Line 429: Please write the name of the reference with proper uppercase letter(s), only at the beginning of the words (excluding “of”, “in”, etc.).
  21. Line 376: Please write the name of the reference with proper uppercase letter(s), only at the beginning of the words (excluding “of”, “in”, etc.).
  22. Line 430: Scientific name must be written in italic, with uppercase letter for the genus.
  23. Line 431: Scientific name must be written in italic, with uppercase letter for the genus.
  24. Line 439: Scientific name must be written in italic, with uppercase letter for the genus.
  25. Line 439: Please write the name of the reference with proper uppercase letter(s), only at the beginning of the words (excluding “of”, “in”, etc.).
  26. Line 441: Scientific name must be written in italic, with uppercase letter for the genus.
  27. Line 442: Please write the name of the reference with proper uppercase letter(s), only at the beginning of the words (excluding “of”, “in”, etc.).
  28. Line 450: Please write the name of the reference with proper uppercase letter(s), only at the beginning of the words (excluding “of”, “in”, etc.).
  29. Line 456: Please write the name of the reference with proper uppercase letter(s), only at the beginning of the words (excluding “of”, “in”, etc.). This is a book, please add the information of the place of the publisher.
  30. Line 457: Please write the name of the reference with proper uppercase letter(s), only at the beginning of the words (excluding “of”, “in”, etc.).
  31. Line 468: Please rewrite the name of the authors, with proper uppercase letter(s), only at the beginning of their name.
  32. Line 468: Please write a complete list of authors. Do not shorten the list of authors using “et al.”

Round 2

Reviewer 2 Report

Congratulations on a great job. The author has made a substantial improvement for this article. The manuscript can be accepted for publishment in the present form.

Reviewer 3 Report

Review of membranes-1557204-v2 
The authors have clarified all issues, this manuscript can be accepted now. 
Note: there are small issues that have to be corrected during the proofreading/post production process, such as:
1. Figure 2: Please revise "seperately" to be "separately"
2. Reference 27: The word after the genus must be written using lowercase letters, such as: zeylanicum, aeruginosa, aureus
3. Reference 30: The word after the genus must be written using lowercase letter --> perforatum